# ActionSense: A Multimodal Dataset and Recording Framework for Human Activities Using Wearable Sensors in a Kitchen Environment

**Joseph DelPreto**[*]**, Chao Liu**[*]**, Yiyue Luo, Michael Foshey, Yunzhu Li,**
**Antonio Torralba, Wojciech Matusik, and Daniela Rus**
MIT CSAIL, Cambridge, MA 02139
{delpreto, chaoliu, yiyueluo, mfoshey, liyunzhu}@csail.mit.edu
{torralba, wojciech, rus}@csail.mit.edu

## Abstract

This paper introduces `ActionSense`, a multimodal dataset and recording framework with an emphasis on wearable sensing in a kitchen environment. It provides rich, synchronized data streams along with ground truth data to facilitate learning pipelines that could extract insights about how humans interact with the physical world during activities of daily living, and help lead to more capable and collaborative robot assistants. The wearable sensing suite captures motion, force, and attention information; it includes eye tracking with a first-person camera, forearm muscle activity sensors, a body-tracking system using 17 inertial sensors, finger-tracking gloves, and custom tactile sensors on the hands that use a matrix of conductive threads. This is coupled with activity labels and with externally-captured data from multiple RGB cameras, a depth camera, and microphones. The specific tasks recorded in `ActionSense` are designed to highlight lower-level physical skills and higher-level scene reasoning or action planning. They include simple object manipulations (e.g., stacking plates), dexterous actions (e.g., peeling or cutting vegetables), and complex action sequences (e.g., setting a table or loading a dishwasher). The resulting dataset and underlying experiment framework are available at https://action-sense.csail.mit.edu. Preliminary networks and analyses explore modality subsets and cross-modal correlations. `ActionSense` aims to support applications including learning from demonstrations, dexterous robot control, cross-modal predictions, and fine-grained action segmentation. It could also help inform the next generation of smart textiles that may one day unobtrusively send rich data streams to in-home collaborative or autonomous robot assistants.

## 1 Introduction

State-of-the-art machine learning for object detection and natural language processing speak to the power of large, high-quality datasets. We seek to facilitate similar leaps forward in the fields of robotic assistants and human action understanding. A key ingredient in training more capable and collaborative robots will be new, high-quality human activity datasets. These must span multiple sensing modalities well-suited to complex, unstandardized environments such as the human home.

Towards this end, we propose `ActionSense`, a new multimodal activity dataset and recording framework with a particular emphasis on wearable sensors. `ActionSense` aims to provide detailed recordings of humans performing everyday actions, enabling insights into manipulation, task planning, and teaching robots by demonstration. It has the potential to advance in-home robotics as well as our

---

[*]These authors contributed equally to this work

36th Conference on Neural Information Processing Systems (NeurIPS 2022) Track on Datasets and Benchmarks.

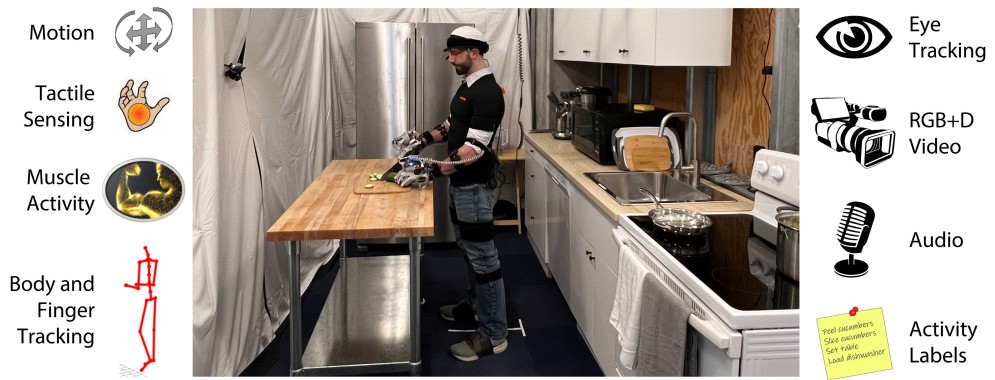

Figure 1: A suite of wearable and external sensors records rich activity information in a mock kitchen.

understanding of how humans perform dexterous manipulations or abstract tasks. As summarized in Figure 1, it uses a multimodal collection of wearable sensors including body trackers, first-person video with attention estimates, muscle activity sensors, and custom tactile gloves made with a matrix of conductive threads. This is coupled with synchronized global perspectives and ground truth data, including activity labels, video from multiple environment cameras, depth camera data, and audio.

Focusing on wearable sensing has multiple benefits. It circumvents issues of occlusions or workspace restrictions, yields fine-grained motion information, and can capture valuable internal states such as muscle activity or attention. Including multiple synchronized modalities enables cross-modal analyses that can probe a person's temporal and spatial reasoning or evaluate which modalities are most useful for certain tasks. Such insights could also help guide the development of future smart textiles. Finally, multimodal wearable sensing is valuable for allowing robots to mimic or assist a person's actions; for example, coupling motion, force, and attention could yield more informative demonstrations and more accurate action predictions.

This paper presents an actively growing human activity dataset, as well as the extensible framework used to capture it. As shown in Figure 1, the dataset is currently focused on daily tasks in a kitchen environment. This is a common target for robot assistants, and therefore one for which `ActionSense`'s insights into human behavior are especially exciting. Moreover, kitchen tasks are structured enough to facilitate extracting commonalities from demonstrations but also abstract enough to introduce high-level reasoning and demonstration variety. Activities are selected to highlight dexterous manipulations, an array of objects and tools, short well-defined tasks, and longer less-structured action sequences.

This project includes the following contributions:

- A framework for recording synchronized multimodal data from wearable and global sensors in a real-world setting, including technical implementations and experimental considerations;

- An expanding dataset focused on daily tasks in a kitchen environment featuring 1) wearable sensors for body and finger tracking, forearm muscle activity, tactile information, and eye tracking with first-person video, 2) synchronized activity labels, environment-mounted color and depth cameras, and microphones, and 3) survey data exploring expertise, workload, and visions for robot assistants;

- Open-source code and instructions for using the presented dataset or recreating the presented framework, which is extensible to additional sensors or environments;

- A plan for long-term retention and availability of the data and code.

The remainder of this paper first places the project in the context of prior work. The `ActionSense` approach, experimental paradigm, sensors, and data management are then summarized. Conclusions and future directions are then discussed. Supplementary materials discuss each topic further.

## 2 Related Work

### 2.1 Activity Datasets

Many large-scale image and video datasets about activities have been presented for action classification, recognition, and description, including Ego4D [8], ActivityNet [10], Charades [33] and its extension to Action Genome [13], Visual Genome [15], and EPIC-KITCHEN-100 [4]. These have amassed large corpuses via, for example, searching online or crowdsourcing. They have yielded successes for understanding human behaviors from the perspective of vision analysis. However, these collection methodologies can also lead to large amounts of variability that may hinder certain types of learning pipelines, and their scale often limits them to focusing on a single modality such as vision.

To help address this issue, multimodal datasets aim to combine vision with additional streams. The NTU RGB+D 120 dataset uses external cameras to provide RGB, depth, body joints, and infrared data during daily activities [19]. CMU-MMAC [5] moves towards integrating wearable devices by recording from external cameras, microphones, five IMUs, and a smart watch during cooking. The MMAct dataset uses external cameras, first-person video, a watch with an IMU, and a smartphone that streams IMU, barometer, and WiFi strength data during daily activities [14]. Others have focused on augmenting first-person video with IMUs [34] or acceleration and heart rate data [23]. The UTD-MHAD dataset combines an RGBD camera with a wearable IMU during gestures or exercise-related motions [3]. A daily intention dataset also featured an on-wrist camera and an accelerometer to observe hand-object interactions [39]. Such datasets explore how multimodal sensing can improve behavior understanding. However, they typically contain a small number of wearable sensors, which restricts the amount of body motion that can be captured, and a small number of modalities, which limits the diversity of behavioral information and context. They also require varying amounts of post-processing to label or parse. Nevertheless, the successes demonstrated by machine learning pipelines built upon them have shown that multi-modal approaches can yield valuable insights.

`ActionSense` aims to leverage a large number of synchronized, labeled data streams to provide detailed activity information during a set of well-defined tasks in a consistent setting. Table 1 presents selected metrics comparing it to previous multimodal datasets. `ActionSense` uses wearable sensors to capture body skeletons, hand poses, attention estimates, muscle activity, and high-resolution tactile sensing. These are coupled with external RGB cameras, depth cameras, microphones, timestamped activity labels, and detailed metadata. Code is provided to parse and analyze the data. These features aim to facilitate learning pipelines that can leverage highly multimodal wearable and global data.

In addition, `ActionSense` provides the software used to stream, save, visualize, and parse the data. Endeavors such as the Microsoft Platform for Situated Intelligence [2] have demonstrated the importance of a flexible framework for online processing of multimodal data for real-time applications. In this vein, `ActionSense` provides its streamlined Python infrastructure for recording from an extensible suite of sensors and managing experiments to create an annotated hierarchical dataset.

### 2.2 Data-Driven Robotics

Data-driven approaches to robotics have been gaining increasing attention due to their potential for flexibility and wide applicability. Past work has demonstrated that even single-task datasets can be highly valuable such as for grasping [27, 16], pushing [6], and poking [1]. Multimodal data can also facilitate fine distinctions, such as material properties of food using vision, audio, and force [30]. Such datasets are useful for tackling particular tasks, but also highlight that collecting real-world robotic training data can be difficult, time-consuming, and costly. This often leads to a lack of diversity

Table 1: Comparison of `ActionSense` to Related Multimodal Datasets

| Dataset | Classes | Subjects | Cameras | Modalities | Environment/Activities | Year |
|---|---|---|---|---|---|---|
| CMU-MMAC [5] | 5 | 43 | 6 | RGB, Audio, 3D Joints, IMU | Kitchen activities | 2010 |
| MSR-Action3D [17] | 20 | 10 | 1 | Depth, 3D Joints | Exercise and arm motions | 2010 |
| UT-Kinect [40] | 10 | 10 | 1 | RGB, Depth, 3D Joints | Daily motions, arm motions | 2012 |
| Multiview 3D Event [37] | 8 | 8 | 3 | RGB, Depth, 3D Joints | Daily actions | 2013 |
| Office Activity [36] | 20 | 10 | 3 | RGB, Depth, 3D Joints | Office activities | 2014 |
| UTD-MHAD [3] | 27 | 8 | 1 | RGB, Depth, 3D Joints, IMU | Exercise and arm motions | 2015 |
| Egocentric Activity [34] | 20 | — | 1 | RGB, IMU | Daily actions, office activities, exercise | 2016 |
| Stanford ECM [23] | 24 | 10 | 1 | RGB, Heart Rate, Accelerometer | Physical activities | 2017 |
| Daily Intention [39] | 34 | 12 | 1 | RGB, Accelerometer | Daily actions | 2017 |
| MMAct [14] | 37 | 40 | 5 | RGB, Smartphone IMU, Smartwatch IMU, Wi-Fi strength, Barometer | Daily actions | 2019 |
| NTU RGB+D [19] | 120 | 106 | 3 | RGB, Depth, IR, 3D Joints | Daily actions | 2020 |
| **ActionSense** | **20** | **10+** | **7** | **19 IMUs, 3D Body Joints, Hand Pose, Gaze, EMG, Tactile, RGB, Depth, Audio** | **Kitchen activities** | **2022** |

that limits applications of embodied intelligence. One way to address this is to collect datasets in simulation. Such environments include virtual human and robotic hands [7], VirtualHome [28], Meta-World [42], and BEHAVIOR [35]. These platforms can reduce the effort needed to generate large amounts of data, but the gap between simulation and reality needs to be bridged.

Alternative approaches include using human supervision to provide diverse high-quality data, which has been the core of many research domains. For example, teleoperation has been used to collect robotic manipulation datasets [32, 21]. These can yield valuable insights, but require careful control of dynamic systems which may be difficult to operate in the real world. Such considerations raise challenges for creating more capable assistants for daily living tasks [38].

In addition to human supervision, learning from human behaviors can augment or simplify data collection. This has benefited applications such as grasping by observing human poses [18], humanoid robot imitation of simple human motions [11] and dancing [24], pouring liquid by using videos of humans from multiple viewpoints [31], or agile quadruped locomotion by imitating a dog [26].

Such approaches demonstrate the value of data-driven robotics and high-quality datasets. However, there is still a gap between existing systems and a future of ubiquitous personal robots. This challenge lies in collecting multi-faceted descriptions of human behavior in structured formats for varied tasks. `ActionSense` aims to take a step towards addressing this by focusing on smart wearable sensing in a kitchen environment and recording labeled multimodal streams during common tasks.

## 3 `ActionSense` Design and Methodology

`ActionSense` aims to support a variety of applications and learning pipelines by providing a dataset and a recording framework for rich multimodal descriptions of human activity in real-world environments. This necessitates considerations such as the scope and diversity of activities, the amount of structure in activities and instructions, the recording paradigm, and the modalities. These are discussed below and in the supplementary materials, to elucidate the dataset and for reproducibility.

### 3.1 Applications and Use Cases

The sensors, activities, protocols, and data were designed to support a variety of future learning pipelines. Applications to robotics include teaching autonomous or collaborative robots by demonstration; this may focus on high-level planning using `ActionSense`'s complex task sequences, or on improving robot control to be closer to human-level dexterity using its more structured activities. These range from sequence prediction models to object grasping and manipulation. Demonstrations may also be used to teach people new skills, particularly by comparing expert and novice performance.

Additional use cases could extract generalizable insights about sensing modalities. For example, cross-modal predictions may use one sensor to predict the output of another sensor; this could eventually reduce the number of sensors in wearable deployments, further probe human motions and object interactions, or explore relationships between egocentric wearable data and third-person video or audio. Multiple modalities could also enable automatic fine-grained action segmentation beyond the provided activity labels, such as using audio or forces to detect individual slices of a cucumber.

Finally, insights extracted from such pipelines could be used to guide future wearable systems. They can inform sensor selection for specific tasks, and guide the development of smart textiles.

While the current work focuses on the dataset and the recording framework, preliminary analyses related to the above envisioned applications are included in the supplementary materials. Activity classification using various subsets of the sensors demonstrates that each modality can provide useful information about the activities on their own, and that using multiple modalities together provides additional insights. A cross-modal correlational analysis also suggests the feasibility of predicting tactile data from muscle activity. Additional models and benchmarks will be explored in the future.

### 3.2 Activities

A paramount consideration of datasets is the selected activities and how they can support the intended use cases. Since `ActionSense` aims to enable insights about how humans manipulate objects or approach household tasks that can be conferred to robots, it features a combination of structured low-level tasks and open-ended action sequences. The current dataset focuses on kitchen tasks; this setting is a common target for assistive robots, provides structure to facilitate comparing iterations across

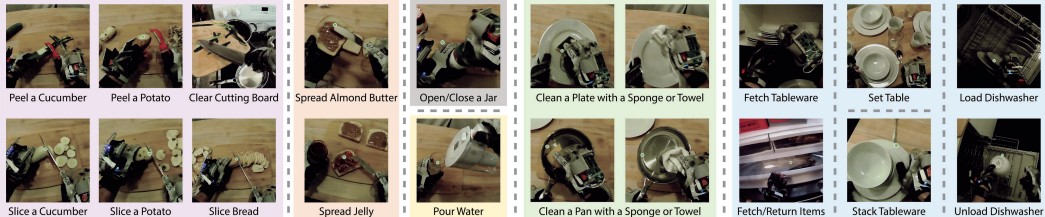

Figure 2: The selected kitchen activities highlight a range of manipulations and action sequences.

subjects, includes high-level planning flexibility, and features dexterous manipulations. Narrowing the scope to improve consistency across subjects helps facilitate a variety of machine learning pipelines to develop more agile, fluid, and dexterous robot controllers. Future expansions of `ActionSense` can include more diverse scenarios to increase its range of applications.

There are currently 20 unique labels in the dataset, as illustrated in Figure 2. This selection is discussed below, where the activities are divided into 6 categories of tasks.

*Peeling and slicing* represent dexterous activities requiring in-hand manipulation with dynamic grasps, coordination between hands, and tool use. They are thus well-suited to a multimodal dataset since aspects such as hand pose, motion paths, force, and attention are all critical. Using cucumbers, potatoes, and bread with disparate hardnesses and shapes precipitates disparate forces, techniques, and tool selection. They can also be performed by both experts and novices, but with different techniques and efficiencies. In addition to these low-level motion and high-level reasoning aspects, the tasks are also interesting for computer vision pipelines since the objects change appearance and subdivide. Finally, they yield an auxiliary labeled activity of clearing the cutting board.

*Spreading* almond butter or jelly on bread uses a knife in a different way. It involves two-handed coordination, varying object appearances, and motions that are repetitive while adapting to the task and object. The consistencies of almond butter and jelly also lead to different techniques.

*Wiping* pans or plates with towels or sponges all aim to clean a flat surface but can have varied approaches. For example, large or small circular or linear periodic motions may all accomplish the goal. The amount of force applied throughout the motion is also a key component. Whether a person, or ultimately a robot, chooses a particular strategy may depend on preference or the object state.

*Opening and closing a jar* are simpler manipulations but still require precise coordination and subtle motions. Tactile forces and muscle activity are also key components of these operations.

*Pouring* water can be informative for prediction or classification pipelines by introducing a transparent liquid that can be hard to model, manipulate, or detect. Each object also continuously changes weight.

*High-level tableware tasks* such as setting a table or loading and unloading the dishwasher introduce more task reasoning. They combine longer sequences of dexterous manipulations with abstracted planning, catering to pipelines that focus on motion primitives as well as action sequence prediction.

All together, these activities span multiple levels of abstraction, manipulations, and objects. Note that cooking tasks such as following a recipe, frying, or baking are left for future extensions; considerations for this decision included focusing on shorter tasks to enable more unique activities and more repetitions, reducing safety risks, and logistics such as installing ventilation and running water.

### 3.3 Experimental Paradigm

The type of instructions, experimental flow, and thoughtful consideration of risks and benefits are vital to implementing effective human subjects experiments that yield impactful data.

During the `ActionSense` experiments, activity orderings and instruction levels are designed to balance efficiency and amount of data variation. Each activity is performed multiple times sequentially before advancing to the next activity; this tends to increase intra-subject consistency, which may be a boon or drawback depending on the learning pipeline. The sequence of activities features shorter well-structured tasks at the beginning and concludes with less-structured planning tasks; this allows subjects to acclimate to the environment and sensors. Finally, instructions describe well-defined objectives while being open-ended enough to introduce varying strategies. For example, table-setting

specifies the number of place settings and the items to use but not how to approach or sequence the task. After receiving instructions, the subject may prefer to proceed through all repetitions of that activity on their own or wait for the experimenter's cues. In all cases, the experimenter watches the subject and uses the recording software to mark when activities start and stop to generate label data.

Care was also taken to ensure safety and privacy. The wearable sensors allow free movement within the predefined space, and commercially available sensors are used when applicable. Subjects are made aware of risks such as from tools, food, or tasks being more awkward than usual when wearing sensors. Subjects are compensated $25 for the approximately 2-hour session, even if they pause or stop at any time. A media release form describes the public dataset including video and audio that may be personally identifiable, and notes that masks can be worn to help obscure identity. The protocol is approved by MIT's Committee on the Use of Humans as Experimental Subjects (COUHES).

### 3.4 Software Framework and Recording Philosophy

The recording software supports synchronized data streaming, saving, and visualizing across multiple sensors with varying sampling rates and data formats. Its class hierarchy streamlines adding new sensors, and multi-threading and multi-processing keep CPU and RAM usage manageable. Post-processing interfaces can also merge streamed data with data recorded by third-party applications.

The software also aims to facilitate higher-quality data with fewer experimental artifacts by allowing researchers to quickly identify issues, record rich metadata without tedious post-processing, and focus more on creating a seamless experience for the subject rather than on implementation details. A GUI allows researchers to label activities in real time, flag activities or calibrations as good or problematic, and submit timestamped notes at any time. The system prints periodic sensor status updates, and streams a customizable composite visualization from multiple sensors as shown in Figure 4.

`ActionSense` uses this framework in conjunction with a philosophy of recording as much continuous data as possible. Not starting and stopping recording between activities provides a more comprehensive view of how humans interact in the kitchen environment, which can be critical for training safe and effective robot assistants. It also enables pipelines that require a complete history of sensor measurements, such as continuous pose estimation from IMUs. Finally, it does not restrict the dataset to the pre-selected activities; researchers may label any "unintentional" activities during post-processing, either manually or automatically via strategic combinations of modalities.

*Synchronizing multiple sensors:* A key aspect of the software is allowing multiple data streams to be synchronized in a flexible and extensible way. `ActionSense` records a wall-clock timestamp for every sample from every sensor. This allows each sensor to operate independently, simplifies online processing, and streamlines adding new sensors. Timestamps may be acquired by the recording software or by device hardware as applicable. Analysis pipelines can use the timestamps to extract portions of data from each sensor that correspond to activity label timestamps. They can also be used to resample streams to unify the number of elements in a given window of time. Supplementary materials include further details about timestamps, synchronizing clocks, and sensor-specific considerations.

*Extensibility:* Given the above synchronization scheme, acquisition from each sensor can occur in its own thread or process without depending on any other sensors. Adding or replacing a sensor does not impact the rest of the pipeline; it merely comprises creating a subclass that provides methods for acquiring samples with appropriate timestamps. Supplementary materials include additional details.

### 3.5 Ethical Considerations

Recording and using human activity data are associated with ethical considerations. Experiments were designed to ensure well-informed consent regarding safety, privacy, and publishing identifiable information. The recruitment pool may be biased for certain metrics. Systems that use the dataset for understanding behavior or training robots should carefully consider concerns such as end-user privacy, dangers of in-home robots, increased worker surveillance using wearables, and job displacement. Such considerations are discussed further in the dataset metadata and supplementary materials.

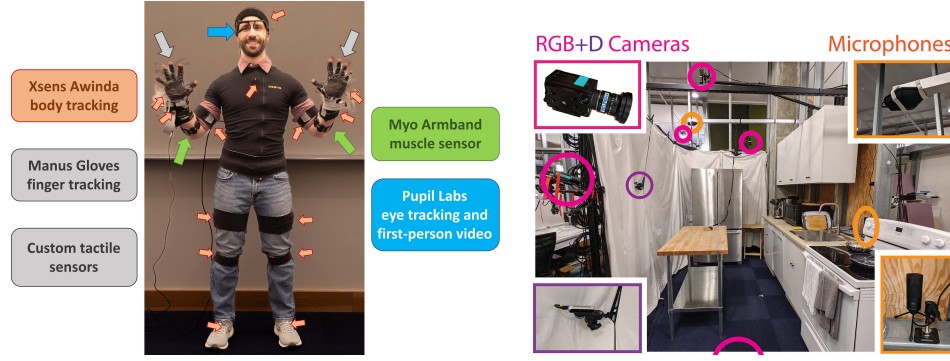

(a) Wearable Sensors       (b) Environment-Mounted Sensors

Figure 3: Synchronized wearable and external sensors provide rich person-centric and global data.

## 4 Wearable Sensors

Figure 3a and Table 2 summarize the current suite of wearable sensors. Focusing on wearables enables fine-grained monitoring of egocentric states or motions. It also avoids workspace limitations, which is valuable for extensions to diverse environments and also for structured settings where dynamic aspects such as fingers or objects may be occluded in third-person views. However, wearing sensors may impact how a person performs tasks and thus affect the studied behaviors; for example, gloves may hinder dexterity. Survey questions focus on the perceived obtrusiveness of each sensor, and the dataset includes some experiments that omit certain sensors to facilitate comparative analyses.

The following sections and the supplementary materials discuss the methods of operation and streaming, third-party software integrations, calibration routines, and limitations of each sensor.

### 4.1 Body and Finger Tracking

The Xsens MTw Awinda system [41] estimates body pose and position. It comprises 17 wireless IMU modules worn on the body using elastic straps, a tight-fitting jacket, and a headband. Custom elastic straps were created to secure the foot sensors rather than using the default shoe inserts. Manus Prime II Xsens gloves [22] augment the skeleton with hand poses by using embedded bend sensors.

The Xsens and Manus applications process data in real time to estimate positions and orientations of body and finger segments. To support future endeavors such as custom calibrations or accuracy evaluations, the continuous `ActionSense` recordings include calibration periods. Calibrations defined by the third-party applications feature known poses. Two custom periods also augment known poses with known global positions: standing at the origin facing along the $x$ axis with arms relaxed (in *N pose*), and standing at $(100\,\mathrm{cm}, 150\,\mathrm{cm})$ facing the opposite direction with arms outstretched (in *T pose*). These are performed at both the start and end of each experiment to help estimate drift.

During experiments, `ActionSense` records data that is streamed from the Xsens software via a network socket. In addition, the Xsens software can record all data in a proprietary format and then reprocess it after the experiment; `ActionSense` includes scripts to import these improved estimates and synchronize them with the rest of the sensor suite. The original data is also provided separately.

*Limitations:* Wearable body and finger tracking facilitates freedom of motion and larger workspaces, but may have lower accuracy than external motion-tracking infrastructure. For example, global positions inferred from the IMUs tend to drift over time. In addition, the Manus gloves can fuse IMUs with bend sensors, but some current experiments disable the IMUs due to observed accuracy issues; this reduces the measured degrees of freedom to finger curling and excludes lateral finger spreading.

### 4.2 Eye Tracking

The Pupil Core headset by Pupil Labs [29] features a wide-angle first-person RGB camera and an infrared camera aimed at the pupil. The Pupil Capture software detects pupil orientations in real time and projects gazes into world-video coordinates to estimate where the person is looking. During calibration, the user gazes at a target while moving their head to vary eye orientation; `ActionSense` records from all sensors to enable re-calibration or accuracy evaluation in post-processing.

Table 2: Wearable Sensors

| Device | Sensor Type | Data Streams | Sampling Rate [Hz] | Calibration Procedures | Third-Party Recording Software |
|---|---|---|---|---|---|
| Xsens Awinda body trackers | IMUs | Body segment orientations and global positions
Joint angles of physical joints and abstract ergonomic joints
Center of mass dynamics
Foot-ground contact points
Sensor accelerations, magnetic fields, and orientations | 60 | Stand with known poses and locations | MVN Analyze |
| Manus Prime II finger-tracking gloves | Bend sensors and optional IMUs | Finger joint angles | 60 | Grasp objects in front of depth camera
Record during Manus Core calibration | Manus Core via MVN Analyze |
| Pupil Labs eye tracker | First-person video
Infrared eye camera | World video with gaze estimation
Pupil detection details (geometry, confidence, etc.) | 30
90 | Record during Pupil Capture calibration | Pupil Capture |
| Myo armbands | EMG

IMU | 8 channels of forearm muscle activity
3-axis acceleration, 3-axis angular velocity
Quaternion pose estimates | 200
50
50 | Stand with known locations and poses | |
| Tactile sensors | Conductive threads on piezoresistive films | 32x32 resistance matrix per hand | 15 | Press on scale while streaming force
Grasp objects in front of depth camera | |

The Pupil Capture software streams data to the `ActionSense` framework via network sockets during experiments, and can also save all data to dedicated files. `ActionSense` includes scripts to merge its recorded data into the main dataset after an experiment, to provide the most consistent sampling rates possible. The original streamed data and the raw recorded data are also available.

*Limitations:* The small wearable sensor provides valuable attention estimates, but it requires a USB cable; `ActionSense` uses a stretchable coiled overhead cable to reduce motion hindrance, but a wearable computer could eliminate this tether. In addition, current experiments use a single eye camera; leveraging binocular information may improve gaze accuracy. Finally, the wide-angle camera is adjusted to view as much of the task as possible, but does not fully span the subject's field of view.

### 4.3 Muscle Activity

A Myo Gesture Control Armband from Thalmic Labs is worn on each forearm. It contains 8 differential pairs of dry EMG electrodes to detect muscle activity, an accelerometer, a gyroscope, and a magnetometer. It also fuses the IMU data to estimate forearm orientation, and classifies a set of five built-in gestures. `ActionSense` wirelessly streams all data by leveraging a Python API [25]. The device normalizes and detrends muscle activity without dedicated calibration, but forearm orientation estimates are relative to an arbitrary starting pose. To facilitate transforming these into global or task reference frames, the calibration poses described in Section 4.1 include known arm orientations.

*Limitations:* The current sensor suite only includes muscle activity from the forearms, which are highly useful for the chosen manipulation tasks but which may not capture all relevant forces and stiffnesses. In addition, the estimated orientations may drift over time.

### 4.4 Tactile Sensors

Custom sensors on each glove provide tactile information. Conductive threads are taped to a pressure-sensitive material that decreases its electrical resistance when force is applied. Threads are oriented perpendicularly to each other on opposite sides of the material, such that each intersection point acts as a pressure sensor. Measuring resistance between each pair of threads yields a matrix of tactile readings. This technique has been explored previously for applications such as smart carpets [20].

The current implementation features 32 threads in each direction that are routed to form a $23 \times 19$ grid on the palm, a $5 \times 9$ grid on the thumb, an $11 \times 4$ grid on the little finger, and a $13 \times 4$ grid on each other finger. A microcontroller and a custom PCB perform processing, multiplexing, and sampling. These are worn on the hand or arm. Data can be streamed via USB or wireless communication.

`ActionSense` provides two types of calibration to facilitate force or pose estimation pipelines. First, the subject presses on a Dymo M25 Postal Scale using a flat hand or individual fingers while weight readings are recorded from the scale via USB. The person then holds 5 unique objects. All sensors are recorded during these activities, including the depth camera and finger joint data.

*Limitations:* These flexible sensors provide high-resolution tactile information, but their sampling rate and material response time may preclude highly dynamic tasks. Calibration periods are designed to help interpret the readings, but accuracy when converting to physical units may vary over time. Finally, the tactile sensors do not reach the fingertips since the underlying gloves feature open fingers.

Table 3: Environment-Mounted Sensors

| Device | Sensor Type | Data Streams | Sampling Rate [Hz] | Calibration Provided |
|---|---|---|---|---|
| FLIR cameras | RGB | Color videos and raw-format frame images | 22 | Intrinsic |
| RealSense depth camera | RGBD | Depth data
Color videos | 15 | Extrinsic |
| Microphones | Omnidirectional, cardioid | Raw audio recordings | 48,000 | |

# 5 Ground Truth, Environment-Mounted Sensors, and Surveys

While the focus on wearable sensors provides detailed activity information and facilitates expanded workspaces, `ActionSense` complements them with environment-mounted sensors and manual labels to provide global perspectives and ground truth for learning purposes. Figure 3b, Table 3, the following sections, and the supplementary materials describe these sensors and software features.

## 5.1 Interactive Activity Labeling

The `ActionSense` software includes a GUI that enables real-time labeling while accommodating unexpected issues or events. It allows researchers to indicate when calibrations or activities start and stop, and to mark each one as good or bad with explanatory notes. Entries are timestamped and saved using the common data format, so pipelines can extract labeled segments from any subset of sensors.

## 5.2 Cameras

*RGB Cameras:* Five FLIR GS2-GE-20S4C-C cameras provide ground-truth vision data of human activities. Four cameras surround the kitchen perimeter, while one camera is mounted above the main table. Images are captured at $22\,\mathrm{Hz}$ with $1600\times1200$ resolution. A custom ROS framework communicates with all cameras. Each camera is calibrated using a checkerboard. The calibration result, raw-format frame images with timestamps, and generated videos are all included in the dataset.

*Depth Camera:* Since most of the chosen activities interact with objects on the table, detailed 3D information of this area can be helpful. An Intel RealSense D415 Depth Camera focused on the table streams data at $15\,\mathrm{Hz}$ with $640\times480$ resolution. An Intel RealSense driver [12] is used to communicate with the camera. Raw images and the depth point clouds are recorded, and extrinsic calibration information is also provided.

*Limitations:* These cameras provide useful ground truth data about the activities in the kitchen environment, but there may be some areas that cannot be viewed while a subject is doing certain tasks (e.g., inside the dishwasher). In such cases, the first-person camera may still provide vision data. In the future, multi-camera calibration can be provided for 3D reconstruction of the environment.

## 5.3 Microphones

Two microphones provide audio information that can be used by learning, segmentation, or auto-labeling pipelines. One has an omnidirectional pickup pattern and is secured above the main table, while the other has a cardioid pickup pattern and is placed on the counter behind the sink. Since the environment is within a common lab space, background noise and speech may be included in the data; this could be a limitation or a benefit depending on the goals of future analysis pipelines.

## 5.4 Survey Data

A post-experiment survey provides metadata and qualitative assessments. These help explore aspects such as how experience levels or handedness influences actions. In addition, it probes how subjects may want to interact with robotic assistants to guide directions of future robotic applications. It also probes their workload and experience with the sensors to guide future wearable development.

# 6 Dataset and Code Repositories: Summary and Accessibility

The `ActionSense` dataset synchronizes streams from all of the above sensors to create a multimodal description of human activities. Composite visualizations as demonstrated in Figure 4a are displayed during experiments, and also included in the dataset so researchers can preview data while developing learning pipelines or labeling. Figure 4b shows the global views captured by RGB and depth cameras.

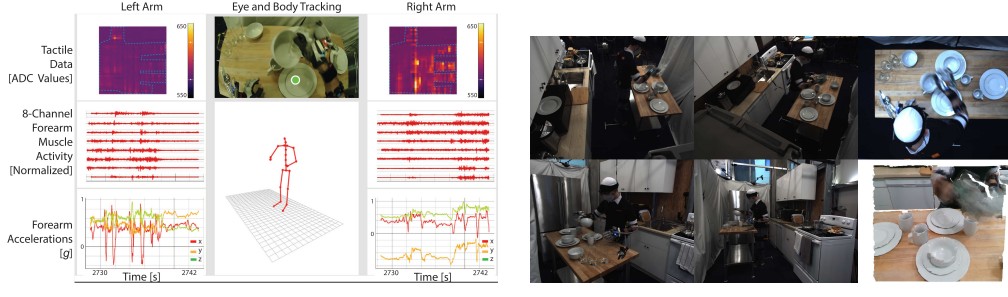

(a) Wearable Sensors                    (b) Global RGB Cameras and Depth Camera

Figure 4: Visualizing wearable and external sensors facilitates real-time or post-processing analysis.

Data is saved in cross-platform formats and organized hierarchically for maximum accessibility and usability. Wearable sensor data leverages the `HDF5` file format [9]. Video and audio data are stored as `AVI` and `WAV` files, respectively, and individual frames are available as raw-format images. Metadata such as body measurements and object measurements are provided as `CSV` files, and future updates plan to add 3D object scans to provide additional manipulation information.

The `ActionSense` code repository includes the Python-based recording framework to stream, visualize, save, and post-process synchronized data. Template classes are included to quickly add new sensors if desired. Explanatory scripts also include examples of analyzing the dataset; they extract sensor streams, segment streams based on labels, and resample streams to unify rates. Code to perform the activity classification and cross-modal analyses in the supplementary materials is also provided. Finally, installation instructions and pre-configured Python environments are included.

All data and code are hosted in repositories with long-term availability. The dataset uses a Creative Commons license, and code uses an open-source license. Data is hosted on MIT servers using storage intended for long-term availability, as is a front-end website at https://action-sense.csail.mit.edu which links to data and code. Third-party repositories may also be used to create archival copies.

## 7    Conclusion

The presented dataset and framework aim to enable learning pipelines that can improve understanding of how humans perform daily activities and improve in-home robot assistants. It focuses on a suite of highly multimodal wearable sensors to capture rich egocentric data while avoiding occlusions and reducing sensing infrastructure. This is coupled with ground-truth labels, external cameras, and microphones to enable, for example, training deep neural networks with various modality subsets.

`ActionSense` uses the framework to record kitchen activities featuring dexterous two-handed manipulations and high-level task planning. This dataset aims to provide detailed synchronized training data in a setting that holds great potential for home automation and collaborative robots. All data, code, and instructions are accessible online in long-term storage repositories.

Future work can use the provided data to create learning and analysis pipelines. These could include cross-modal evaluations, fine-grained action segmentations, sequence predictions, dexterous robot assistants, and comparisons of expert and novice techniques. The dataset continues to grow, and including more subjects or more activities will continue to increase the utility of the dataset. Alternative wearable sensors or new modalities could also be explored to address limitations of the current collection. Future work can also use the provided software to record multimodal data from additional sensors or in additional settings.

In these ways, `ActionSense` strives to take a step towards enabling more capable robot assistants and guiding the next generation of smart wearable devices.

## Acknowledgments and Disclosure of Funding

This work was supported by the GIST-MIT Research Collaboration grant funded by the Gwangju Institute of Science and Technology (GIST) in 2021-2022. The authors declare no conflicts of interest. The authors are also grateful for the insights of James Bern during the paper-writing process.

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
