# OpenReview forum: "ActionSense: A Multimodal Dataset and Recording Framework for Human Activities Using Wearable Sensors in a Kitchen Environment"
_NeurIPS.cc/2022/Track/Datasets_and_Benchmarks — NeurIPS 2022 Datasets and Benchmarks _

### Official Review · Reviewer_hqyL · 2022-07-15
**A Rich Action Dataset with Detailed Documentation**

**Rating:** 8
**Confidence:** 4
**Correctness:** I don't have concerns on the correctn…
**Clarity:** The paper is very clear and easy to f…

**Strengths:**

- A wide range of sensors and activities to provide a very rich dataset.
- The website is very well documented with clear instructions on different sensors/activities with corresponding data streamings.
- The topic can be potentially interesting to a wide range of audience.

**Weaknesses:**

- It is unclear how much impact brought by different sensors, especially gloves, on participants' natural behavior of cutting, peeling, etc. However, in order to collect the dataset, such an effect seems to be inevitable. I would recommend the authors mention it in the limitation.

- Why doesn't the activity task include some representative cooking tasks (e.g., frying, stirring, grilling, and baking, to name just a few). Please justify.

- The number of participants is never mentioned in the main text. From the website, the current dataset has 6 subjects, which is limited. However, with the pipeline established, it is not difficult to collect data from more participants. As the website mentioned, they aim for 25 subjects by 2022 Fall.

- The paper may benefit from some initial discussion of potential usage of the dataset

**Additional Feedback:**

I want to thank the authors for doing a great job! I am looking forward to seeing the camera-ready version of the paper!

**Documentation:**

The dataset is documented very well.

**Ethics:**

The paper discussed the ethical concerns appropriately.

**Relation To Prior Work:**

Another egocentric dataset Ego4D: Grauman, Kristen, Andrew Westbury, Eugene Byrne, Zachary Chavis, Antonino Furnari, Rohit Girdhar, Jackson Hamburger et al. "Ego4d: Around the world in 3,000 hours of egocentric video." In Proceedings of the IEEE/CVF Conference on Computer Vision and Pattern Recognition, pp. 18995-19012. 2022.

**Summary And Contributions:**

In this paper, the authors collected a dataset with 6 subjects (25 as planned by the time it is published) on performing various kitchen cooking tasks. A number of sensor streams are captured, including motion, eye tracking, tactile sensing, EMG, body and finger tracking, audio, video, together with activity labels and subjective surveys. Subjects did a series of activities, including peeling and slicing, spreading, opening/closing a jar, wiping, pouring, and high-level tableware tasks. The authors envision that the potential usage of the dataset includes task planning, robots training, and smart textiles guiding. I appreciate the authors creating such a rich dataset with excellent documentation and decent web pages.

---

> ### Author Response · Authors · 2022-08-15
> **Reply to comments**
>
> Thank you for your thoughtful review, we appreciate your time and valuable feedback.  We are currently working on revisions that address your questions, and wanted to pass along some initial comments below.  We also look forward to any additional notes or advice that you may have.
>
> - It is a great point that wearing the sensors will inevitably impact how subjects perform various tasks.  For example, the gloves may affect dexterity for fine-grained manipulations.  The survey includes a few questions about the obtrusiveness of each sensor, and we hope to glean some insights from this as the dataset continues to grow.  For the revision, we plan to update the paper sections about sensor limitations and general discussions to ensure they clearly mention that behaviors may be affected by wearing sensors.
>
> - As you mention, the current suite of activities omits typical cooking tasks such as frying or baking.  We agree that these would be valuable to add in the future.  For the current dataset though, a few of the considerations that lead to the decision to omit them are below:
>   - Cooking tasks can be time-consuming, and would thus significantly limit the number of activities (and repetitions) that we could collect.
>   - Additional safety concerns for the subjects, and possible sources of damage for the sensors.
>   - Logistical and infrastructure considerations: the mock kitchen does not currently feature running water, functional heating elements, or adequate ventilation and other safety features for cooking.
>
> In light of these aspects, we decided to focus on shorter tasks that would allow for a wide range of activities to be explored (and for multiple repetitions of most activities).  They still cover many common kitchen and manipulation tasks, and are thus still useful for the target applications.  This also allows us to isolate various interesting elements such as manipulation-focused activities or planning-focused activities.
>
> We will plan to describe these considerations more in the main paper and the supplemental material, since they are important for the experiment design and dataset usage as you suggest.
>
> - We agree that additional discussion of potential dataset usages would be beneficial, and will plan on adding a section on this in the main paper.  Some example applications include teaching robots by demonstrations, controlling robots to be closer to human-level dexterity, cross-modal predictions (e.g. predicting tactile information from muscle activity - see the supplemental material for an initial demonstration of feasibility), fine-grained action segmentation using multiple modalities, high-level task planning, and guiding smart textiles.
>
> - Thank you for passing along the reference to Ego4D; this is an exciting and highly relevant dataset.  We are aware of it, and will add it to the related work discussion.

---

### Official Review · Reviewer_VF6M · 2022-07-24
**ActionNet: A Multimodal Dataset for Human Activities Using Wearable Sensors in a Kitchen Environment**

**Rating:** 6
**Confidence:** 4
**Clarity:** The paper is clearly written and easy…

**Strengths:**

1. Compared to other existing activity datasets, ActionNet contains way more modalities so that it is able to capture more detailed and comprehensive information about human interaction with the environment and have a better description of human activities.

2. The dataset itself is well-organized and easily accessible. The authors provide detailed information and guidance about the information of the dataset and the user guide, which benefits the potential users.

3. The dataset may be valuable for the potential development of smart textiles and the emulation and assistance of robots for a person's action.

**Weaknesses:**

1. The size and diversity of the ActionNet are limited compared to existing activity datasets. The ActionNet currently contains 10 subjects where each subject performs around no more than 20 activities with about 78.5 minutes of videos on average. However, the existing activity dataset, e.g. ActivityNet, contains 648 hours of videos and 200 classes, which is far more than the ActionNet. The authors claim ActionNet circumvents the issue of workspace restriction (line 38) while all videos are taken in the same mock kitchen. The paper will be much stronger if it contains more diverse scarioes and activities.

2. Lack of experiments and applications. The paper does not contain any applications or experiments that can be benefit from this dataset, which undermines the utility of this paper. Also, since the dataset contains a large amount of multimodal information, it is better to show ablation studies that how each modality is useful compared to only vision in existing activity datasets. It may raise my concern that some modalities may not be that useful in understanding human activities.

**Additional Feedback:**

No additional feedbacks.

**Correctness:**

Most claims made in this paper are correct while some of them may need further evaluation, as mentioned in weakness.

**Documentation:**

The paper includes sufficient domumentation for the dataset.

**Ethics:**

No ethics issues found in this paper.

**Relation To Prior Work:**

The authors carefully and comprehensively place their work upon existing activity datasets.

**Summary And Contributions:**

This paper presents a new multimodal dataset ActionNet which records the activities of human activities in a mock kitchen environment with multiple sensors. The dataset contains multiple tasks including object manipulations, dexterous actions, and complex action sequences that are commonly seen in kitchen-related activities and all tasks are labeled by authors. The paper utilizes a large number of sensors to record the activities from the aspects of vision, sound, motion activities, and tactile feelings using wearable and external sensors.

---

> ### Author Response · Authors · 2022-08-15
> **Reply to comments**
>
> Thank you for your thoughtful review, we appreciate your valuable feedback and aim to incorporate it into the revised version of our paper.  A few comments as we prepare the updates are below.  We also look forward to any additional comments or suggestions you may have.
>
> - While the size of the dataset is smaller than others such as ActivityNet, we aim to focus on providing a rich multi-modal description of well-defined actions, with synchronized and well-organized data streams from wearable and global sensors.  This can facilitate applications ranging from teaching robots by demonstration to cross-modal analysis.  While endeavors such as ActivityNet that collect videos from the internet are highly valuable, they focus on a single modality (video, typically third-person) and have a large amount of variation between examples (disparate environments, large differences between examples of an activity, variable recording quality, differing vantage points, moving cameras, etc).  They thus aim to facilitate different usages and pipelines than the current proposed dataset.  Thank you for highlighting this and providing the opportunity to elaborate.
>
> - We agree that diversifying the data and including more scenarios would be beneficial, and we are excited about doing so in the future.  For the current dataset though, we have chosen to focus on everyday kitchen tasks.  This is a common target for assistive robots, provides a relatively structured scenario that enables comparing iterations across subjects, provides high-level planning flexibility, and includes dexterous manipulations.  Narrowing the scope to improve consistency across subjects helps facilitate a variety of machine learning pipelines to develop more agile, fluid, and dexterous robot controllers.  The presented framework and wearable sensors can be used in more diverse scenarios though, and we look forward to doing so in the future to continue expanding ActionNet.
>
> - Using ActionNet to create machine learning pipelines for a variety of applications is certainly an exciting path forward that we are currently exploring.  Applications include teaching robots by demonstrations, controlling robots to be closer to human-level dexterity, cross-modal predictions (e.g. predicting tactile information from muscle activity - see the supplemental material for an initial demonstration of feasibility), fine-grained action segmentation using multiple modalities, high-level task planning, and guiding smart textiles. We plan to add some discussion to the paper that summarizes possible use cases. For the current work though, we have focused on the dataset itself and the extensible recording framework. We believe these can be valuable for researchers in a variety of contexts, and are thus eager to make their availability known.

---

> > ### Comment · Reviewer_VF6M · 2022-08-29
> > **Thank you for your response!**
> >
> > I would like to thank the authors for their thorough responses. I have no further questions.

---

### Official Review · Reviewer_1uXC · 2022-07-26
**Review of ActionNet: A Multimodal Dataset for Human Activities Using Wearable Sensors in a Kitchen Environment**

**Rating:** 5
**Confidence:** 3
**Correctness:** The dataset is constructed correctly.
**Clarity:** The paper is without an application p…

**Strengths:**

The paper is the first to provide a multimodal dataset and recording framework with an emphasis on wearable sensing in a kitchen environment. It uses motion, tactile sensing, body and eye tracking, audio and image to form a dataset of kitchen activities.

**Weaknesses:**

3.1: The paper presents a human activity dataset with the extensible framework used to capture it. However, the selected activity in the paper is very limited (6 activities). The author claims it is a growing dataset but having 6 activities is not enough for a large dataset to produce robust results, as claimed in the Introduction.
There is a lack of user application discussion in the paper. In particular, what use cases can be beneficial from this paper. Many kitchen activities require more information than a camera and motion, for example, temperature, shape, the texture of the different objects, etc. The author needs to provide an application that justifies the use case of what is captured in the dataset.

**Additional Feedback:**

N/A

**Documentation:**

The dataset is well documented.

**Ethics:**

I don't have ethical concerns.

**Relation To Prior Work:**

Relation to prior works is well discussed.

**Summary And Contributions:**

The paper presents a multimodal dataset and recording framework with an emphasis on wearable sensing in a kitchen environment. It uses a wearable sensing suite to capture motion, force, and attention information, coupled with different action sequences in the kitchen.

---

> ### Author Response · Authors · 2022-08-15
> **Reply to comments**
>
> Thank you for your thoughtful review, we appreciate your time and valuable feedback.  We agree that addressing these points can help to strengthen the paper, and are currently working on revisions accordingly.  We wanted to pass along the below comments in the meantime, and also look forward to any additional notes you may have.
>
> - We agree that having a large number of activities helps to strengthen a dataset and expand its applicability.  The current version of ActionNet features 20 labeled activities, in addition to labeled calibration procedures (including known hand poses, object grasps, tactile forces, etc.).  Note that section 3.1 summarizes these activities using 6 categories; each category contains multiple labeled tasks.  We will update Figure 2 and the text of section 3.1 to convey this more clearly - thank you for pointing this out, as it is an important metric for the dataset.
>
> - Thank you for also providing the opportunity to expand upon the envisioned use cases, since as you mention it is important to understand what the dataset can benefit.  We will plan on adding a section on this in the main paper; some example applications include teaching robots by demonstrations, controlling robots to be closer to human-level dexterity, cross-modal predictions (e.g. predicting tactile information from muscle activity), fine-grained action segmentation using multiple modalities, high-level task planning, and guiding smart textiles.
>
> - It’s a good point that some common kitchen tasks would benefit from additional sensors as well, such as temperature or more information about the manipulated objects.  We can consider adding these in the future to expand the range of possible applications, but we feel that the current suite of sensors offers a rich description of the activities and enables a variety of usages.  We can elaborate on these design decisions in the paper, their associated limitations, and the benefits of future additional sensors.
>   - As an additional project supported by the dataset, we are working with collaborators to create a virtual environment based on the ActionNet data and setting.  Part of this includes scanning the manipulated objects, so we can offer these models for download alongside the data in the future to provide more texture and shape information.
>   - As a note regarding temperature sensing, many of the tasks that would benefit most from this modality involve heating.  Such cooking activities are currently omitted from the dataset, but if they are included in the future then adding temperature sensing could certainly be helpful.  Note that the response to reviewer hqyL has more information about the design decision to omit cooking tasks.

---

### Official Review · Reviewer_Ju3w · 2022-07-28
**A multimodal dataset and synchronized multimodal data recording framework**

**Rating:** 5
**Confidence:** 4

**Strengths:**

- The authors developed a framework to record multimodal sensor data synchronously.
- The author collected human activity from multiple diverse wearable and sensor data.

**Weaknesses:**

- As the authors claim that one of the contributions of this work is to develop a synchronized multimodal data recording system, I would expect how it is different from the existing multimodal data collection system, such as PSI(Platform for Situated Intelligence: https://github.com/microsoft/psi). What is the novelty of this data collection framework compared to the existing data collection frameworks?
- Additionally, I missed the details of their proposed synchronized data collection framework. More specifically, I would like to know more about the following things:
```
-  How the framework collects data from multiple sensors synchronously? For example, how can data be collected synchronously from RGBD, Pupil Core headset, and wearable sensors? How are these sensors synchronized?
- How the synchronization works? Is it offline (i.e., data are collected from multiple sensors independently and then in the post-processing step [after the completion of data collection] the data are synchronized a) or online [data are collected synchronously, and there is no need for post-procession to synchronize the data streams after the data collection]? Please explain concretely.
- How are the online annotations performed? I would like to suggest the author prepare a short video demo of how the data labeling system works.
- During the online annotations: are the participants continuously performing the sequence of actions without any interruption from the experimenter? or an experimenter guides the participants on when to start and end one action and start another?
```
- Author claims this framework "........ is extensible to additional sensors or environments" [Line 61]. I would like to know how the researchers can extend this framework to include additional sensors. For example, how the framework ensures the synchronized data collection for the additional sensor. How easy will the extension be? Do the researchers need to change almost all the components of this framework to include additional sensors?
- The author provides the list of libraries to recreate the environment to execute the data collection framework. However, I would suggest that the author share a docker to make it easy and accessible for other researchers to recreate the environment easily.
Several multimodal activity datasets are available in the literature; how does this new dataset contribute differently to the research community? I suggest the author prepare a table to compare this new dataset to the existing and recent multimodal activity datasets, such as UTD-MHAD, MMAct, NTU-RGBD, etc.


**Additional Feedback:**

Please check my previous comments.

**Clarity:**

The paper is well written. But I found some missing details based on the aims of this paper. It seems to me that one of the primary contributions of this paper is the data collection framework. If it is then I would expect more concrete details about this data collection framework [please check my comments in the weakness section]

**Correctness:**

Some of the claims are sound. I have some questions and confusion about some of the other claims which I stated in the weakness sections.

**Documentation:**

I could able the download the dataset and there is some documentation to install and test the data collection framework. As I stated I would suggest the author share a docker to make the data collection framework accessible and easy to recreate the environment. Additionally, they did not find the details of synchronized data collection and how can the researcher extend this framework to include additional sensors or even change the existing sensors. For example, how can the researchers replace the depth camera from Intel RealSense with another related device?

**Relation To Prior Work:**

I would like to know how this data collection framework differs from the existing framework. and how the collected multimodal datasets address the lacking of the existing multimodal datasets. I would like to suggest the author compare this dataset to the recent multimodal activity datasets, such as MMAct.

**Summary And Contributions:**

The authors present a framework to record multimodal data from multiple wearable and visual sensors. Using this framework they collected human activity dataset. The data samples contain a sequence of activities with multimodal sensor data.

---

> ### Author Response · Authors · 2022-08-15
> **Reply to comments**
>
> Thank you for your valuable feedback, we appreciate your thorough review. We are currently updating the paper to incorporate the suggestions, and believe that they will help strengthen the work. In the meantime, we wanted to pass along a few comments below. We look forward to any additional remarks or advice that you may have.
> - Thank you for highlighting the additional related works including Microsoft’s PSI, MMAct, UTD-MHAD, and NTU-RGBD. We will incorporate these into the related work section, and also make a comparison table.
> - We appreciate the opportunity to expand on technical details such as synchronization, extensibility, and annotations; as you mention, these are crucial aspects of the presented system. We plan on adding summaries in the main paper, and expanded dedicated sections in the supplemental materials. The website will also be updated to highlight these aspects and new descriptions. Some remarks on them in the meantime are included below.
> - Sensors are synchronized by recording a wall-clock timestamp for every sample from every sensor during acquisition. This aims to provide a flexible synchronization method; each sensor can operate independently, it facilitates extensibility, simplifies online processing demands, and avoids intensive offline processing.
>   - For each sensor, a vector of timestamps is stored alongside the list of data matrices. Offline analyses such as machine learning pipelines can use these vectors to extract data from all sensors for a specific activity.
>   - Two possible concerns with this approach are a) the time delay between when a sensor measures a sample and when Python ingests the sample, and b) synchronizing clocks.
>     - In all cases, a timestamp is recorded when Python receives the data. This unifies the process across sensors, and is always applicable regardless of whether the sensor has an onboard clock. This streamlines adding new sensors.
>     - If a sensor has an onboard clock, this is recorded in the dataset as a separate time vector for enhanced accuracy.
>     - We use one computer for all wearable sensors, and another for environment-mounted sensors. The former computer uses the latter computer as its time server to synchronize wall clocks.
>   - A few notes on specific sensors:
>     - Xsens records timestamps on-device. These are merged with streamed data in post-processing for increased accuracy.
>     - The Pupil Core headset also has its own clock. The ActionNet code synchronizes it with the computer’s clock at startup.
>     - FLIR cameras provide per-frame timestamps.
>     - The Myos and tactile sensors do not have on-board clocks, so their timestamps reflect when the data was received. The Bluetooth and Serial delays are hopefully negligible for most applications.
>     - Microphone data is buffered and provided to Python in chunks. The sampling rate is 48kHz and the chunk size is 2048 samples, so timestamps are recorded at ~23Hz.
> - Including a video of the online annotations on the website is a great idea, and we will do this shortly. As a summary, the code creates a GUI where the experimenter can select the upcoming activity from a drop-down list. They then press a button to start the activity, and a button to stop the activity. Once stopped, they can rate the activity as good or bad and enter any notes (for example, if something unexpected happened and the label should be discarded).
>   - Wall-clock timestamps are recorded when they start and stop the activity.
> - Before a new set of activities, the experimenter provides instructions then tells the subject to start whenever they are ready. For the repetitions of each task, such as peeling 3 cucumbers, some subjects prefer iterating on their own while others prefer waiting for explicit cues. In both cases, the experimenter watches the subject to press the start/stop buttons.
> - To extend the framework to a new sensor, a new subclass can simply be created that inherits from SensorStreamer. It should provide a method that acquires samples and timestamps, then passes them to the parent class. All other functionality is inherited automatically (data logging, visualization, organization within the HDF5 hierarchy, etc.).
>   - No parts of the framework need to be updated except for adding a new subclass.
>   - We may actually switch to a Kinect instead of the RealSense camera; this will just involve writing the interface code to acquire frames and timestamps.
>   - Some of our collaborators have actually added their own sensors already, including shoe sensors and a smart watch. These classes are now in the GitHub repository.
> - Note that the ActionNet code currently includes template files for using the timestamps to extract labeled segments from all sensors, and a template class for adding a new sensor.
> - We agree that providing a pre-configured environment would make installation much easier; we plan to create either a Docker or Conda environment that can be downloaded with the code. Thank you for this suggestion.

---

> > ### Comment · Reviewer_Ju3w · 2022-08-29
> > **Final ratings**
> >
> > Thanks for the response. Although this work has good potential to contribute to the multimodal machine learning domain, the contributions of the current version are minimal compared to the recent multimodal datasets, such as MMAct. Although I raised this concern regarding the novelty of this work, I found the author did not address that comprehensively. I would suggest the author incorporate a diverse set of environments instead of using a fixed kitchen environment. Also, I agree with Reviewer MWR4, I would change the dataset name, as this dataset only contains the kitchen environment. Based on the authors' response, I am inclined to keep my previous rating.

---

### Official Review · Reviewer_MWR4 · 2022-07-30
**Review of ActionNet: A Multimodal Dataset for Human Activities Using Wearable Sensors in a Kitchen Environment**

**Rating:** 6
**Confidence:** 5
**Correctness:** The paper is correct for the most part.
**Clarity:** The paper is clear and well written.

**Strengths:**

The paper is excellent in that it simultaneously measures wearable sensors in addition to the environment-mounted sensing that has been done in action recognition to date in order to achieve a better understanding of human daily activities and a learning pipeline to improve in-home robotic assistants.

The fact that it targets kitchen activities that include two-handed operation and high-level task planning as tasks is also considered a valid future application.

**Weaknesses:**

The name of the dataset is not appropriate. The name of the dataset is limited to kitchen activities, but it is misleading to think that it covers ActionNet and general activities.

The superiority of this dataset should be indicated precisely in the comparison table with the dataset described in the Related Work section. The number of sensors, the number of recorded actions, and the number of samples do not clearly indicate the superiority of this dataset.

As a publisher of the dataset, shouldn't you benchmark the dataset with a basic model and indicate the baseline in the dataset? When the dataset is cited in other papers, it will be difficult to verify the performance, and the validity of the dataset will be reduced.

**Additional Feedback:**

In the case that both hands are involved, it may be more effective to take not only the Tactile sensor but also the hand-pose in order to divert the approach to the object. However, since simultaneous measurement seems to be difficult, it is not included in the Weakness.
If you consider this in the future, the effectiveness of this dataset will be enhanced.

**Documentation:**

It seems to be well documented in https://action-net.csail.mit.edu/

**Ethics:**

No ethics issues

**Relation To Prior Work:**

Isn't Action Genome more appropriate than Visual Genome [11] as a prior literature on action recognition?

Action Genome: Actions as Composition of Spatio-temporal Scene Graphs, CVPR2020
https://www.actiongenome.org/

**Summary And Contributions:**

This paper proposes a framework for recording synchronized multimodal data from wearable and global sensors in the real world to enable a learning pipeline to better understand human everyday behavior and improve in-home robotic assistants.

In addition to making the data publicly available, open-source code and instructions for reproducing the presented framework are available.

---

> ### Author Response · Authors · 2022-08-15
> **Reply to comments**
>
> Thank you for your insightful review, we appreciate your time and valuable feedback.  We are currently preparing revisions that aim to incorporate your comments to improve the paper, and wanted to pass along a few responses in the meantime.  We look forward to any additional comments or suggestions you may have.
>
> - Thank you for the suggestions and comments on the related work; we are planning to add the citations and generally create a clearer comparison including a table.
>
> - It’s a good point that the name ActionNet conveys a general tone while the current dataset is limited to kitchen tasks.  Our objective is to collect data about a wide range of tasks so we can teach robots how to perform activities more effectively and more similarly to humans.  We started with the everyday tasks in the kitchen, and we can ultimately rename this subset of ActionNet as it grows to reflect this focus.
>
> - Using ActionNet to create a variety of machine learning pipelines and benchmarks is certainly an exciting path forward that we are currently pursuing.  Applications include teaching robots by demonstrations, controlling robots to be closer to human-level dexterity, cross-modal predictions (e.g. predicting tactile information from muscle activity - see the supplemental material for an initial demonstration of feasibility), fine-grained action segmentation using multiple modalities, high-level task planning, and guiding smart textiles.  We plan to add discussion to the paper that summarizes possible use cases.  For the current work, we have focused on the dataset itself and the extensible recording framework. We believe these can be valuable for researchers in a variety of contexts, even if they are decoupled from an initial model trained on the data, and are thus eager to make their availability known.
>
> - We also agree that recording hand pose alongside tactile data can be highly informative, and your comments about using this for two-handed manipulations are particularly intriguing.  Any further thoughts you may have on this direction would be appreciated.  Note that the current suite of sensors includes the Manus finger-tracking gloves, onto which the tactile sensors have been mounted, and the Xsens body trackers.  This thus provides synchronized streams of hand poses, relative positions of both hands, and force data.  Most activities are also performed within the field of view of an RGBD camera, in addition to the multiple RGB cameras, which can provide additional pose information.

---

### Author Response · Authors · 2022-08-26
**Revision Uploaded**

We would like to thank all reviewers again for their thoughtful and constructive feedback.  We have uploaded a revised version of the manuscript that aims to address all of the comments, and think that they have helped to improve the paper.  We look forward to any additional comments that you may have.

The revisions include adding discussions that help to clarify envisioned applications, current contributions, and technical/experimental aspects.  They also augment the related work section to better situate ActionNet within the context of existing datasets and frameworks.  The paper also now includes an initial machine learning pipeline to perform activity classification, along with modality ablation studies; these aim to demonstrate an application for the dataset and explore the value of multiple modalities.

A more detailed summary of the revisions is included below.   A copy of the manuscript with blue text highlighting the updated portions is also included in the supplementary materials ZIP file.

*Related Work*
- Updated Section 2 to include the suggested citations and to better situate ActionNet within the context of these related datasets.
- Table S3 in Section A.3 presents metrics that compare ActionNet with other multimodal datasets.

*Discussions and Limitations*
- Added Section 3.1 to describe envisioned use cases and applications.  It also mentions conclusions from the preliminary activity classification, modality ablations, and cross-modal analyses now included in the supplementary materials.
- Section 3.2 now describes rationale behind narrowing the current scope to kitchen tasks, and mentions future plans to expand ActionNet to more diverse scenarios.
- Section 3.2 now explicitly states that there are 20 activity labels included in the dataset.  Figure 2 is also updated to show this more clearly, and to denote the 6 categories discussed in the text.
- The end of Section 3.2 addresses the omission of cooking tasks.
- Section 4 now includes a discussion of wearable sensors affecting how subjects perform tasks.
- Section 4 also now mentions that avoiding occlusions can be helpful even in structured environments such as the mock kitchen.

*Technical and Experimental*
- Section 3.3 describes how experimenters provide start/stop cues to the subjects.  Supplementary section C.4 adds more detail.
- Section 3.4 adds a subsection about how sensor streams are synchronized.  Supplementary section E.2.1 describes this in detail.
- Section 3.4 adds a subsection about how sensors can be added or replaced without impacting the rest of the pipeline.  Supplementary section E.2.2 describes this in more detail.
- Section 6 mentions that pre-configured Python environments will be provided (using either Docker or Conda).
- Section 6 describes the sample processing scripts and template classes that facilitate analysis and extensibility.
- Section 6 mentions that additional object metadata will be included in the dataset, including measurements and eventually 3D scans that may include texture.
- Extended Supplementary section E.4 to elaborate on how the GUI allows experimenters to annotate activities in real time.  It also details how the generated timestamps are stored and used for segmentation.  A video will be added to the website shortly.
- Added text in Supplementary section E.3 to describe sample scripts and template classes that are included in the code repository to facilitate synchronized segmentation, analysis, and sensor suite extensions.

*Preliminary Machine Learning Pipeline and Results*
- Performed activity classification and modality ablation studies to demonstrate an application of the dataset and to show the value of the multi-modal approach.
  - It includes a data processing and segmentation pipeline, a neural network architecture, a leave-one-subject-out cross validation training and evaluation, and a discussion of the results.
  - This complements the existing cross-modal analysis that explores the feasibility of predicting tactile data from muscle activity.
- These are detailed in Supplementary sections F and G, and Section 3.1 summarizes conclusions.

*Website and Dataset Stats*
- Updated dataset statistics and survey results in the Supplementary sections A and H to reflect new experiments.  There are now 10 subjects in the dataset.  The target is still approximately 25 subjects.
- The website will soon offer Conda and/or Docker environments to streamline setup.
- The website will soon include a video of the GUI including the labeling process.
- The website will soon offer object measurements, and eventually 3D object scans.

---

### Meta-Review · Area_Chair_g19D · 2022-09-04

**Recommendation:** Accept
**Confidence:** 3

**Metareview:**

This paper presents an interesting multimodal dataset for research work on analyzing human actions. While there are some issues with this work, the authors have addressed most of them well during the rebuttal phase. Overall, there is sufficient support to accept this paper at this conference. The authors should include all these points in the revised paper and dataset before the conference date.

---

### Decision · Program_Chairs · 2022-09-16

Accept